# Seroprevalence of *Strongyloides stercolaris* in Patients about to Receive Immunosuppressive Treatment in Gran Canaria (Spain)

**DOI:** 10.3390/tropicalmed8030181

**Published:** 2023-03-20

**Authors:** Cristina Carranza-Rodríguez, Laura López-Delgado, Álvaro Granados-Magan, José-Luis Pérez-Arellano

**Affiliations:** 1Unidad de Enfermedades Infecciosas y Medicina Tropical, Hospital Universitario Insular de Gran Canaria, 35016 Las Palmas de Gran Canaria, Spain; 2Departamento de Ciencias Médicas y Quirúrgicas, Universidad de Las Palmas de Gran Canaria, 35016 Las Palmas de Gran Canaria, Spain

**Keywords:** *Strongyloides stercoralis*, kidney transplantation, biological therapies

## Abstract

*Strongyloides stercoralis* infection is generally asymptomatic or mildly symptomatic, but in the immunosuppressed host, it is associated with more severe and complicated forms with a worse prognosis. *S. stercoralis* seroprevalence was studied in 256 patients before receiving immunosuppressive treatment (before kidney transplantation or starting biological treatments). As a control group, serum bank data of 642 individuals representative of the population of the Canary Islands were retrospectively analyzed. To avoid false positives due to cross-reactivity with other similar helminth antigens present in the study area, IgG antibodies to *Toxocara* spp. and *Echinococcus* spp. were evaluated in cases positive for *Strongyloides*. The data show this is a prevalent infection: 1.1% of the Canarian population, 2.38% of Canarian individuals awaiting organ transplants and 4.8% of individuals about to start biological agents. On the other hand, strongyloidiasis can remain asymptomatic (as observed in our study population). There are no indirect data, such as country of origin or eosinophilia, to help raise suspicion of the disease. In summary, our study suggests that screening for *S. stercoralis* infection should be performed in patients who receive immunosuppressive treatment for solid organ transplantation or biological agents, in line with previous publications.

## 1. Introduction

Strongyloidiasis is a disease caused by different species of soil-transmitted helminths of the genus *Strongyloides* [1]. *Strongyloides stercoralis* is the main causative agent, although other pathogenic species have been described in specific regions, such as *S. fuelleborni fuelleborni* in Africa or *S. fuelleborni kellyi* in New Guinea. The estimated prevalence of infection ranges from 30 to 100 million people worldwide, although there are indirect data which suggests that these values may be underestimated [2,3,4]. The highest prevalence is in Southeast Asia, Sub-Saharan Africa, and Latin America. Although Europe is not considered an endemic area, this is debated in the case of Spain [5,6].

*S. stercoralis* most commonly infects humans through the cutaneous route. Filariform larvae (L3) in the soil infect the host by penetrating intact skin Filariform larvae (L3) can be found in the soil and penetrate intact skin. Other less frequent routes of infection are person-to-person transmission after receiving an organ transplant, oro-anal practices or contact with secretions during hiperinfection syndrome. The infectious capacity leads to autoinfection, which is the main difference between the life cycle of *S. stercoralis* and other soil-transmitted helminths. Filariform larvae generated by the host can invade intestinal mucosa (endogenous autoinfection) or perianal skin (exogenous autoinfection) and restart the infectious cycle. Autoinfection has important consequences: it is responsible for the persistence of the parasite for years (even decades) after infection, and larval penetration can lead to dissemination to other organs and/or transport of intestinal bacteria into the bloodstream.

Clinical manifestations of strongyloidiasis are very diverse [7,8]. Acute infection is rarely observed, and symptoms (cutaneous, respiratory, and digestives) are usually mild. Chronic infection is more frequent and, although asymptomatic in a high percentage of cases, it can also present with cutaneous, respiratory, and digestive symptoms. There are two forms of severe strongyloidiasis: the disseminated form and hyperinfection syndrome. Disseminated strongyloidiasis develops when larvae penetrate organs and tissues other than those of its traditional life cycle (skin, lungs, and gastrointestinal tract). Hyperinfection syndrome develops with the appearance of conditions that facilitate parasite development and its access to the bloodstream. Disruption of the intestinal membrane predisposes to bacteraemia caused by intestinal microbiota and the transport of these bacteria to other tissues, such as the lungs or the central nervous system, which can lead to sepsis, pneumonia, or meningitis. The main findings in complementary studies are eosinophilia and increased plasma IgE concentration that suggests the Th2 response to the helminth. Eosinophilia is fluctuant in chronic forms, and its absence, therefore, does not rule out infection. Furthermore, in complicated forms, eosinophilia tends to disappear, which makes diagnosis even more challenging.

A change in immune status can increase the number of parasites, leading to severe forms of strongyloidiasis. The increased number of people receiving organ transplants and biological agents in recent years means more patients are at risk of infectious diseases. However, the prevalence of *S. stercoralis* in patients about to receive immunosuppressive treatment is not well documented. Furthermore, these patients are not systematically screened for its detection. It is important, therefore, to know the prevalence of strongyloidiasis among patients who will receive immunosuppressive treatment and to determine the best screening strategies for early diagnosis and treatment of strongyloidiasis before immunosuppression to minimize the likelihood of progression to severe forms of the disease.

Our main objective was to determine the prevalence of *S stercoralis* antibodies in patients about to receive immunosuppressive treatment (specifically before kidney transplants or biological treatments).

## 2. Materials and Methods

### 2.1. Study Site

The study was conducted at the Hospital Universitario Insular de Gran Canaria (HUIGC), Canary Islands. Spain. It is a public hospital that serves a mixed population (rural and urban) of approximately 500,000 people on the island of Gran Canaria. The Canary Islands is an archipelago located in the Atlantic Ocean. It is about 100 km from Morocco and the western Sahara and about 1400 km from the Iberian Peninsula (Figure 1). The Archipelago has mild thermal conditions, an average temperature of 22 °C, and mild winter and pleasant summer temperatures.

### 2.2. Ethics Approval Statement

The present study followed the guidelines of the Declaration of Helsinki in 1975, revised in 2013. All procedures performed in this study were approved by the Institutional Ethics Committee (ref 2021-208-1) prior to the start of the study. A signed informed consent was obtained from every participating patient before they participated in the study, and patients were completely anonymized by the researchers. The researchers followed every mandatory (health and safety) procedure.

### 2.3. Study Design and Pupulation

A prospective study was conducted between January 2019 and March 2020 among all patients attending the immunoprophylaxis outpatient clinic of the HUIGC. Overall, 270 patients attended, and 256 met the inclusion criteria for this study. All patients over fourteen years of age (>14 years) evaluated before receiving immunosuppressive treatment (including patients before kidney transplants or before starting biological treatments) were included in the study. Fourteen patients were excluded from the study because of sample insufficiency, incorrect serum elution o refused to participate.

For the control group, 642 serum samples from the Canary Islands population were used (Table 1). The data was collected as part of a study for the Canary Nutrition Survey. The study population consisted of the entire population of the Canary Islands registered in the census between 5 and 75 years of age. The selection of individuals was made through a two-phase cluster study, with the municipality as the first variable and the individual sample as the second variable.

### 2.4. Variables

The following data was collected from patients in the study group:(i)*Patient demographics*: sex (male and female), age, country of origin (Spain, rest of Europe, Africa, America, and Asia), place of residence and occupation.(ii)*Type of patient*: pre-biological treatment or pre-kidney transplant(iii)Underlying pathology(iv)*Presence of eosinophilia*, defined as a total eosinophil count ≥450/µL and relative eosinophilia, defined as the percentage of eosinophils >5% but eosinophil count < 450/µL

In the control group, age, sex, and island of residence were obtained for each subject.

### 2.5. Serological Study

Approximately 5 mL of blood sample was collected by venipuncture in a plain vacutainer. The serum obtained by centrifugation of these samples was stored at −80 °C before being tested. For serology testing, a commercial qualitative immunoenzymatic method based on the enzyme-linked immunosorbent assay (ELISA) technique (DRG Instruments, Marburg, Germany) was used for the determination of specific immunoglobulin G (IgG/IgM) antibodies, following the manufacturer’s recommendations. The diagnostic specificity of this assay is 94.12% (95% confidence interval: 83.76–98.77%), and the diagnostic sensitivity is 89.47% (95% confidence interval: 75.2–97.06%).

Briefly, the serum samples were diluted 1:100 with a phosphate buffer solution and pipetted into the corresponding wells, coated with a soluble fraction of *S. stercoralis* L3 filariform larval antigen [9], leaving the first well empty to be used as a blank for the study. After this, the plate strips were sealed with the supplied self-adhesives and incubated for one hour at 37 °C. The microplates are coated with specific antigens that bind to the antibodies in the sample. After washing the wells to remove all unbound sample material, horseradish peroxidase (HRP) conjugate was added. This conjugate binds to the captured antibodies, with the unbound conjugate being removed in a second wash step. Next, a tetramethylbenzidine (TMB) solution was pipetted, which allowed visualization of immune complex formation by blue staining produced in positive samples due to the enzymatic reaction that occurs. Subsequently, a sulfuric acid solution was pipetted as a stop solution into the wells, thus stopping the reaction and causing a color change from blue to yellow. From the use of controls (positive, negative and cut-off) supplied with the kit, it was possible to validate the assay using a microplate reader (photometer) interpreting the extinction with an absorbance reading of 450/620 nm. The results were read as a function of the extinction value (OD) obtained in the sample, divided by the cut-off point (cut-off) and multiplied by 10, expressed in DU units (DRG^®^ particular units). A positive result was determined if the value obtained was greater than 11 units (DU) and negative below 9 DU. Between 9–11 was in the “intermediate zone”.

To avoid false positives due to cross-reactivity with other similar helminth antigens present in our study area, anti-*Toxocara canis* and *Echinococcus granulosus* IgG antibodies were detected. Novalisa^®^ (NovaTec Immunodiagnostica GmbH, Hessen, Germany) with >95% sensitivity and specificity was used to measure serum anti-*Toxocara canis* IgG antibodies, according to the manufacturer’s protocol. In brief, all samples were diluted 1:100 with IgG sample diluent, and all controls (*T. canis* IgG-positive, *T. canis* IgG-negative, *T. canis* IgG cutoff, and substrate blank) were prepared. The following requirements must be satisfied for an assay to be considered valid: cutoff was 0.150–1.300, negative controls were <cutoff, positive controls were >cutoff, and the substrate blank was <0.100. For interpretation, the results were calculated to NovaTec units (NTU), samples with >11 NTU were considered positive. However, if the NTU value was between 9 and 11, the sample was considered equivocal, and a fresh sample was repeated. If the results of the repeated test were also equivocal, the sample was considered negative. ELISA-detecting *E. granulosus*-specific IgG antibodies (DRG, Marburg, Germany) were used. The test was used in serum samples according to the manufacturer’s recommendations. Ten IU was taken as the threshold value and values exceeding it were considered positive. They were evaluated in cases positive for Strongyloides.

### 2.6. Data Analysis

Descriptive statistics were used to summarize demographic data and clinical characteristics. Continuous variables are presented as means and standard deviation (SD) or median and range (when SD was >50% of mean). Categorical variables were summarized as absolute numbers or frequencies (percentages) and analyzed using appropriate tests. Data were analyzed with SPSS software for Mac (version 25.0, SPSS Inc., Chicago, IL, USA).

## 3. Results

### 3.1. Study Group

Of the 256 patients, 144 (56.2%) were women, and 112 were men (34.8%). Ages ranged from 14 to 83 years old and were distributed normally, with a mean age of 46 ± 15. Regarding the origin of the patients, 231 (90.2%) were from Spain, while the remaining 25 were foreigners from four continents: America (10), Europe (6), Africa (6) and Asia (3). None of the patients was agricultural or farm workers, and the vast majority (30%) were employed in the service sector.

Patients were classified into two groups: (a) awaiting kidney transplant (48) and (b) with immunological/inflammatory conditions requiring immunosuppressive treatment (208). The most frequent diagnoses in this group are shown in Table 2.

Eosinophilia, the main laboratory finding in patients with strongyloidosis, was encountered in twenty-one patients (8%), and 44 (17%) had relative eosinophilia (Table 3). None of these patients had IgG antibodies to *Strongyloides* spp.

Twelve patients (4.7%) tested positive for IgG antibodies to *S. stercoralis*, and the rest were negative. The twelve positive cases were tested for cross-reactivity with other helminths (*Toxocara canis* and *Echinococcus granulosus*), and no antibodies were detected. The sex distribution showed no statistical differences. With respect to age distribution, nine (75%) of the 12 patients belonged to the age group classified as young [14–40 years]. For the origin of patients with positive serology, 10 of the 12 patients (83.4%) were from Spain, while the remaining two (16.6%) were foreigners and came from Africa (Morocco and Equatorial Guinea). Ten (83.3%) of the 12 patients with a positive serology were about to receive immunosuppressive or immunomodulatory therapy, and the other two patients (16.7%) were on haemodialysis awaiting kidney transplants. The characteristics of patients with positive results are shown in Table 4.

### 3.2. Control Group

A serological study was performed on 642 samples: 358 were women (55.6%), and 284 were men (44.4%). Ages ranged from 5 to 75 years old; the mean age was 38 ± 19.6. After analysis of the serological results, seven positive results were obtained (1.1.%). Of the positive results, the majority (71.4%) were females, and three cases were under 18. None of the cases was in the older age group (>65 years). No antibodies to Toxocara canis or Echinococcus granulosus were detected in any of the cases (Table 5).

## 4. Discussion

*S. stercoralis* infection is generally asymptomatic or shows mild symptoms of the disease. In the immunosuppressed, however, it is associated with more severe, complicated forms with a worse prognosis.

Knowing the prevalence of *S. stercoralis* infection in a given population is very important, particularly if individuals are about to undergo immunosuppressive therapy. However, the information on global prevalence provides very variable results. It depends on the number of studies carried out in individual countries, the type of population evaluated, and the diagnostic method used [4,10].

The diagnosis of strongyloidiasis is problematic since multiple techniques are available, but they all have limitations, so there is no true “gold standard” [11,12,13]. Conventional diagnosis of strongyloidiasis is based on different techniques that allow for *finding larvae in biological samples*. Direct visualization of larvae in stool samples is the simplest technique (Ritchie’s technique). Other diagnostic methods are based on larvae migration because of their thermotropism (Harada-Mori or Baermann), faecal charcoal culture (that maintains the pH of the medium and provides a medium in which the larvae can develop and allow them to develop to the filariform stage) or the visualization of bacteria dragged by the movement of larvae in agar cultures. All these techniques are very specific but have several drawbacks.

On the one hand, they require the use of fresh non-refrigerated samples and the study of feces obtained on different days (usually three samples). On the other hand, in chronic forms, the number of larvae is scarce, and their elimination is intermittent, so their sensitivity is low. Of all the techniques mentioned, agar culture has the highest sensitivity. Other direct techniques have been described, such as *genetic material detection* in different biological samples, the most common being detection in feces [14,15,16,17,18,19,20]. In general, the sensitivity of PCR is lower than that of previously described techniques with a highly variable false negative rate: 12% [18], 20% [20], 39% [15] and 50% [17]. There are two main reasons why PCR in feces presents lower sensitivity: (i) the smaller amount of the sample compared to direct identification techniques and (ii) the thickness of the larvae wall, which can be difficult to detect parasite DNA [20]. However, the increase in the faecal sample is associated with the frequent presence of PCR inhibitors in DNA extracts. On the other hand, parasitic lysis techniques only slightly improve sensitivity [20]. Finally, most of the techniques used, except for some more recent ones, are not commercially accessible, which limits their usefulness [20].

For this reason, the meta-analysis published by Buonfrate in 2018 [19] concludes that PCR might not be suitable for screening purposes, whereas it might have a role as a confirmatory test. Multiple serological techniques have been described for the diagnosis of strongyloidiasis that differ in several aspects: (i) the applied methodology; (ii) the immunoglobulin isotype detected, and (iii) the type of antigen used [21,22,23,24,25,26]. In different studies, the applied methodology is variable (i.e., enzyme-linked immunosorbent assay [ELISA], dipstick methods, luciferase immune-precipitation systems [LIPS] or immunofluorescence antibody test [IFAT] [22,25]. Although the specificity of LIPS seems superior to that of the other techniques [21,25], ELISA is the most widespread technique in serological diagnosis, thanks to its simplicity and the possibility of automation [26]. The humoral response to *Strongyloides stercoralis* infection includes the generation of different immunoglobulin isotypes (IgM, IgA, IgG1, IgG4, and IgE). The generation of IgM or IgA declines quickly and, therefore, does not have great diagnostic value in infected persons except in the initial acute phase (4–6 weeks) [24].

Furthermore, the IgA response in serum is lower than in other secretions (i.e., saliva) [24]. The measurement of IgE anti-Strongyloides *stercoralis* has been interpreted as an early marker and a measure of active infection, although the information is limited [24]. For this reason, IgG measurement is the main serological test in chronic strongyloidiasis, used in most studies. The utility of the specific determination of IgG1 (depressed in older populations) or IgG4 (consistently found in chronically infected) does not provide additional data [24,26]. In many serological studies, non-commercially available antigens are used, which limits their use reproducibility among different batches [20]. For some years, there have been marketed techniques that use two types of antigens: crude or recombinant [25,26]. The use of recombinant antigens is more specific but, in general, less sensitive. On the contrary, the techniques that use raw antigens are more sensitive but less specific, so it is necessary to rule out other helminth infections [26]. The sensitivity of the technique increases if the cut-off point is high and decreases in immunocompromised patients, which may reflect a decreased level of antibody production in this population [13].

Considering the objective of this study, it was necessary to know the prevalence of infection by *Strongyloides stercoralis* in our environment. In Spain, although most of the reported patient series correspond to cases imported from endemic areas, autochthonous cases of strongyloidiasis have also been documented in isolated cases and in series. The prevalence of autochthonous infection in Spain has not yet been fully determined and varies according to previous series. In our study, the prevalence in the Canary Islands is 1.1%, slightly higher than in previous Spanish reports [27]. For this reason, in this study, we used serology, considering that it is the best screening strategy suggested by Carnino et al. [23] in subjects waiting for immunosuppression in a country with a low prevalence (<5%) [4]. Specifically, we used a commercial technique with a crude antigen due to its higher sensitivity. To evaluate the false positive cases due to cross-reactivity, in the cases with *Strongyloides* spp. Positive serology, the presence of antibodies against two helminths in our environment (*Echinococcus* spp. and *Toxocara* spp.), was determined to be negative in all of them. The presence of other geohelminths (*Ascaris* spp., *Trichuris trichura* or hookworm) was not evaluated due to the anecdotal presence of these parasites in the native population in our environment [28].

Use of steroids [29], HTLV-I infection [30] and HIV infection [31] are forms of immunosuppression known to favour the development of complicated forms of strongyloidiasis. More recently, two other immunosuppressive states have been added to the list of factors linked to severe strongyloidiasis: (i) solid organ transplant [32] and (ii) use of biological agents.

With respect to solid organ transplants, two main mechanisms can lead to the development of strongyloidiasis in organ recipients. Firstly, the transmission of parasites through the donated organ has been documented [33]. Secondly, using immunosuppressive agents (including steroids) to prevent organ rejection can increase the parasite load in previously infected patients [34,35]. Hence international recommendations include screening for *S. stercoralis* infections in transplant donors and recipients, specifically if they are from endemic areas and/or have eosinophilia [36,37]. In our study, 2 of 48 (4.7%) organ recipients were found to have IgG antibodies to *S. stercoralis*, one of whom (2.38%) was Spanish. In the literature, the prevalence of infection in this type of patient in a non-endemic area range from 3 to 12.4% [36,38,39,40].

The development of biological agents represents a breakthrough in treating many autoimmune diseases and malignancies. These agents act by inhibiting components of the normal immune response, such as circulating molecules (cytokines, immunoglobulins), cellular receptors or intracellular proteins [41]. However, there is little information on strongyloidiasis in patients undergoing biological treatment. Most of the published cases are associated with the use of anti-TNF therapy (etanercept [42], infliximab [43], adalimumab [44] or golimumab [45], and attributed to the accumulation of circulating CD4 T helper type 1 lymphocytes, with a corresponding decrease in T helper type 2 cells [46]. A smaller number of cases has been reported in patients receiving monoclonal antibodies against IL-6R (tocilizumab) [47], anti-CD20 (rituximab) [48] or anti-CD52 (alentuzumab) [49]. It should be noted that these patients also received different doses of steroids, and one also had HTLV-I infection, so a clear causality cannot be conclusively established. In the group of patients about to receive biological agents, there are several interesting findings in this study: (i) 10 of the 208 (4.8%) had IgG/IgM to *S. stercoralis*. Similar data have also been reported in other series in different geographical locations [50,51,52]. (ii) Two-thirds of these patients had an underlying pathology whose therapeutic options included an anti-TNF agent. (iii) Only one of the patients was a foreigner (not from Spain). (iv) None of these patients had absolute or relative eosinophilia.

Our study has some limitations. Thus, there is a temporal difference between the control and study groups, which could explain some of the differences found. However, due to the improvement in socio-sanitary conditions, a lower prevalence could be expected and not the opposite. On the other hand, this is an initial study, so only a serological technique was used. Considering the results obtained, it will be necessary to complement them with direct study techniques in future studies.

In summary, our study suggests that patients receiving immunosuppressive treatments for solid organ transplants or biological agents should be screened for *S. stercoralis* infection, in line with previous publications [23]. The data that justify screening (modified from [53]) are: (i) It is a prevalent disease: 1.1% of the Canarian population, 2.38% of Canarian individuals awaiting an organ transplant, and 4.8% of individuals about to start a biological agent; (ii) strongyloidiasis can remain asymptomatic (as observed in our study population), and there are no indirect data, such as country of origin or eosinophilia, to help raise suspicion of the disease; (iii) the test we used has high sensitivity and specificity, and is easily accessible and reproducible; (iv) a simple and effective therapeutic agent is available (ivermectin) with few side effects, short duration, and low cost; (v) early diagnosis and treatment will minimize the likelihood of disease progression to severe forms and complications.

## Figures and Tables

**Figure 1 tropicalmed-08-00181-f001:**
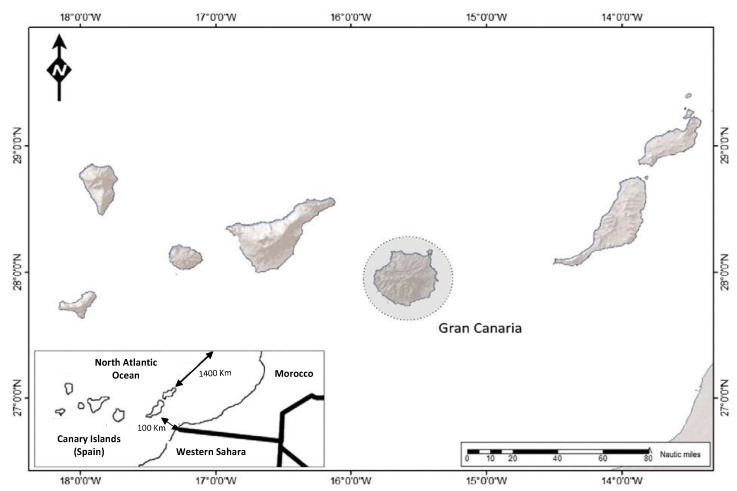
Map of the Canary Islands. Inset: the minimum distances from West Africa and mainland Spain are shown.

**Table 1 tropicalmed-08-00181-t001:** Distribution of the control group samples by Islands.

Island	Women	Men	N (%)
Gran Canaria	142	109	251 (39.1%)
Tenerife	97	75	172 (26.8%)
La Palma	37	44	81 (12.6%)
La Gomera	22	22	44 (6.9%)
Lanzarote	23	10	33 (5.1%)
Fuerteventura	20	11	31 (4.8%)
El Hierro	17	13	30 (4.7%)
Total	358	284	642

**Table 2 tropicalmed-08-00181-t002:** Main previous diseases of patients in the study group.

Baseline Pathology	N = 208
Crohn’s disease	57 (27.5%)
Psoriasis	45 (21.6%)
Multiple sclerosis	40 (19.2%)
Rheumatoid arthritis	16 (7.7%)
Ulcerative colitis	12 (5.8%)
Hidradenitis	10 (4.8%)
Ankylosing spondylitis	7 (3.4%)
Atopic dermatitis	3 (1.5%)
Uveitis	3 (1.5%)
Others	15 (7.2%)

**Table 3 tropicalmed-08-00181-t003:** Eosinophilia values in the study group.

	Immunosuppressive Treatment (204) N (%)	Kidney Transplant (48) N (%)
Relative eosinophilia		
<5%	173/204 (85.0)	35/48 (73.0)
>5%	31/204 (15.0)	13/48 (27.0)
Absolute Eosinophilia		
<450	189/204 (92.6)	42/48 (87,5)
>450	15/204 (7.4)	6/48 (12.5)

**Table 4 tropicalmed-08-00181-t004:** Characteristics of patients with *Strongyloides*-positive serology (N = 12).

	Age	Sex	Country of Birth	Baseline Pathology	Eosinophilia (Absolute and Relative)	*S. stercoralis* IgG (DU)	*Toxocara canis*	*Echinococcus granulosus*
Case 1	40	Female	Spain	Crohn’s disease	No	14	Negative	Negative
Case 2	32	Male	Spain	Multiple sclerosis	No	23	Negative	Negative
Case 3	70	Male	Morocco	Lung cancer	No	12	Negative	Negative
Case 4	35	Male	Spain	Crohn’s disease	No	17	Negative	Negative
Case 5	37	Female	Equatorial Guinea	Kidney transplant	No	21	Negative	Negative
Case 6	31	Female	Spain	Crohn’s disease	No	15	Negative	Negative
Case 7	40	Male	Spain	Crohn’s disease	No	117	Negative	Negative
Case 8	37	Male	Spain	Psoriasis	No	12	Negative	Negative
Case 9	37	Female	Spain	Multiple sclerosis	No	67	Negative	Negative
Case 10	29	Female	Spain	Crohn’s disease	No	18	Negative	Negative
Case 11	62	Female	Spain	Hydradenitis	No	55	Negative	Negative
Case 12	74	Female	Spain	Kidney transplant	No	68	Negative	Negative

DU: Results in Units, Positive >11, Indeterminate 9–11, Negative < 9.

**Table 5 tropicalmed-08-00181-t005:** Characteristics of patients with *Strongyloides*-positive serology (N = 7).

	Age	Sex	Country of Birth	Eosinophilia (Absolute and Relative)	*S. stercoralis* IgG (DU)	*Toxocara canis*	*Echinococcus granulosus*
Case 1	13	Female	Spain	No	20	Negative	Negative
Case 2	15	Female	Spain	No	18	Negative	Negative
Case 3	17	Male	Spain	No	35	Negative	Negative
Case 4	41	Male	Spain	No	22	Negative	Negative
Case 5	43	Female	Spain	No	14	Negative	Negative
Case 6	57	Female	Spain	No	45	Negative	Negative
Case 7	62	Female	Spain	No	18	Negative	Negative

DU: Results in Units, Positive >11, Indeterminate 9–11, Negative < 9.

## Data Availability

The dataset used for this study is available from the corresponding author.

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
