# Peer review of "Seroprevalence of Strongyloides stercoralis in Patients about to Receive Immunosuppressive Treatment in Gran Canaria (Spain)"

_tropicalmed, 2023, doi:10.3390/tropicalmed8030181_

Round 1

Reviewer 1 Report

The manuscript describes a seroprevalence study of Strongyloides infection in patients about to receive immunosuppressive treatment in the Canary Islands.  Several concerns need to be addressed:

1. Please describe how the sample size was determined.

2. To check for cross-reactivity to other helminth infections, why were only antibodies to  Toxocara spp and Echinococcus spp tested? What about antibodies to  STH such as Ascaris, Trichuris, and hookworm- are these not endemic in the study site? 

3. Do you think other commercial Strongyloides ELISA kits (eg. Bordier, NovaLisa) may produce different results? Is there a need to compare the results obtained with a few kits?

4. Is there a need to perform real-time PCR on stool samples of seropositive patients to confirm the Strongyloides infection? Please discuss.

5. Describe the limitations of the study.

6. Add recommendations for the best screening strategies in the population. 

Reviewer 2 Report

- It is better to mention the name of the region or place of study in the title.

- Considering the importance of strongyloidiasis  in immunosupprssed individuals, why didn't authors evaluate IgM antibody, which indicates an active infection.

- It has been well documented that the gold standard diagnostic technique for strongyloidiasis is the agar plate culture(APC) method. Why did the authors not perform this test in addition to serology?
